# Comparison of Biosafety and Diagnostic Utility of Biosample Collection Cards

**DOI:** 10.3390/v14112392

**Published:** 2022-10-29

**Authors:** Hanna Keck, Michael Eschbaumer, Martin Beer, Bernd Hoffmann

**Affiliations:** Institute of Diagnostic Virology, Friedrich-Loeffler-Institut, Federal Research Institute for Animal Health, Suedufer 10, 17493 Greifswald-Insel Riems, Germany

**Keywords:** biosample collection cards, FTA (Flinders Technology Associates) cards, Ahlstrom-Munksjö, Whatman, Copan, Macherey-Nagel, virus isolation, real-time RT-qPCR, sequencing, transfection, Lipofectamine 3000

## Abstract

Six different biosample collection cards, often collectively referred to as FTA (Flinders Technology Associates) cards, were compared for their ability to inactivate viruses and stabilize viral nucleic acid for molecular testing. The cards were tested with bluetongue virus, foot-and-mouth disease virus (FMDV), small ruminant morbillivirus (peste des petits ruminants virus), and lumpy skin disease virus (LSDV), encompassing non-enveloped and enveloped representatives of viruses with double-stranded and single-stranded RNA genomes, as well as an enveloped DNA virus. The cards were loaded with virus-containing cell culture supernatant and tested after one day, one week, and one month. The inactivation of the RNA viruses was successful for the majority of the cards and filters. Most of them completely inactivated the viruses within one day or one week at the latest, but the inactivation of LSDV presented a greater challenge. Three of the six cards inactivated LSDV within one day, but the others did not achieve this even after an incubation period of 30 days. Differences between the cards were also evident in the stabilization of nucleic acid. The amount of detectable viral genome on the cards remained approximately constant for all viruses and cards over an incubation period of one month. With some cards, however, a bigger loss of detectable nucleic acid compared with a directly extracted sample was observed. Using FMDV, it was confirmed that the material applied to the cards was sufficiently conserved to allow detailed molecular characterization by sequencing. Furthermore, it was possible to successfully recover infectious FMDV by chemical transfection from some cards, confirming the preservation of full-length RNAs.

## 1. Introduction

The use of untreated cotton-based filter paper for mass screening of newborns for inherited metabolic diseases in the 1960s [1] was one of the first steps towards biosample collection cards. During the following years, the principle of blood samples dried on fiber-based material was also applied to other investigations [2,3]. This method allowed the use of smaller sample volumes for the detection of pathogens and space-saving storage at room temperature over a long period of time [4,5]. For some viruses, desiccation alone already has an inactivating effect [3], but for others this is not the case [6]. For this reason, additional coatings of the cellulose material have been established. In most cases these consist of chaotropic or anionic substances [7] and are able to lyse white blood cells [8], tumor cells [9], most types of bacteria [10,11], or viruses [12,13,14,15] and to denature proteins [16]. Nucleic acid storage cards that are treated with inactivating substances are often collectively referred to as Flinders Technology Associates (FTA) cards after the institution where they were originally invented in the 1980s, Flinders University in Adelaide, Australia [17]. Once the sample is applied to the card, the structure of the pathogen is disrupted and the nucleic acid is released [18]. The fibers of the matrix then capture the free nucleic acid and preserve it [16,19,20]. Additionally, the now immobilized genetic material is protected by the coating substances from further degradation by nucleases, oxidases, or from UV damage [16,19,21,22]. As a result, a stabilized sample is obtained that does not need to be refrigerated [20,22] and can be shipped in a standard envelope via letter mail [13,23,24]. 

Originally developed for long-time preservation of DNA [1,22], the principle of the cards is now used for DNA and RNA in almost all analytical fields [25], including virology [5,7,26,27,28], bacteriology [29], parasitology [11,30], monitoring of mosquito-borne pathogens [31], genomic analyses [32,33], pharmacogenetics [34], forensics [35], and bio-banking [18]. The array of sample matrices is correspondingly wide, ranging from body fluids to fresh tissue samples of humans, animals, or plants [26,36,37]. DNA stored on cards at room temperature was successfully amplified by PCR after several years [9,19,36,37]. Available data on the stability of RNA on cards usually cover several months [5,13,15,16,21], but in some cases only a few weeks [22,38]. Over these time spans, preserved RNA was found to be relatively resistant against the influence of temperature and humidity [12,20,22]. The utility of the recoverable RNA depends on the sample material, the type of nucleic acid (single-stranded or double-stranded), the storage duration, and the sensitivity of the assay used [38]. If RNA samples are to be stored for a long time, refrigeration to 4 °C or lower is recommended. [16]. At temperatures of −20 °C or below, RNA samples on cards are stable for over a year [39].

The preserved samples are suitable for virological and serological analyses [40], but further characterization or culturing of the pathogens is constrained, because the direct isolation of infectious virus from the card is not possible [13,14,22,23]. For viruses with a single-stranded positive-sense RNA (+ssRNA) genome, however, the intact viral genome by itself is capable of replication. If the virus genome is introduced into permissive cells, it functions as an mRNA [41], which results in replication and ultimately the generation of new infectious virus particles. In the laboratory, this can be achieved by different transfection methods [42,43].

Previously published data were almost exclusively collected with the “FTA Classic” card by Whatman (now part of Cytiva, Marlborough, MA, USA). Meanwhile, there are many other manufacturers that also offer cards with various coatings for diverse use cases. In this paper, we use “biosample collection card” (BCC) as a generic term for all cards. We tested five other BCCs in addition to the FTA Classic card. Two standard filter papers, which are not intended for sampling, were also included to see if they are a viable alternative when no BCCs are available. Virus inactivation and nucleic acid stabilization were analyzed at different time points for a wide variety of viruses: bluetongue virus (BTV), family *Reoviridae*, double-stranded RNA genome, no envelope [44]; foot-and-mouth disease virus (FMDV), family *Picornaviridae*, single-stranded RNA genome, positive polarity, no envelope [45]; small ruminant morbillivirus (also known as peste des petits ruminants virus, PPRV), family *Paramyxoviridae*, single-stranded RNA genome, negative polarity, lipid envelope [46]; and lumpy skin disease virus (LSDV), family *Poxviridae*, double-stranded DNA genome, lipid envelope [47]. All are highly relevant animal pathogens that cause notifiable transboundary diseases.

## 2. Materials and Methods

### 2.1. Viruses and Cells

BTV-5 RSArrrr/05, FMDV A_22_/IRQ/24/64, PPRV Nigeria 75/1, and LSDV Neethling V100 were used for the stabilization and inactivation experiments. Two additional FMDV isolates (A/IRN/08/2015 and O/FRA/1/2001) were used for the sequencing and transfection experiments. The cell lines and media for virus culture are listed in Table 1. For FMDV, BHK-21 cells were used for propagation and subsequent titration and LFBK-αVβ6 cells were used for the inactivation, stabilization, and transfection experiments.

For continuous culture, the cells were propagated in the appropriate medium supplemented with 10% fetal bovine serum (FBS). For the preparation of a master stock per virus, the cells were infected as monolayers in 75 cm^2^ cell culture flasks (Corning, NY, USA) without using FBS. After 2 h, medium with 10% FBS was added in equal amounts to reach a final serum concentration of 5%. This was followed by an incubation at 37 °C. For BTV, infected cultures were incubated for up to 5 days, for FMDV 3 days, for PPRV 4 days, and for LSDV 6 days. The optimal time for virus harvest was determined by close observation of the development of cytopathic effect (CPE). 

For FMDV, the culture supernatant was clarified by centrifugation for 1 min at 2000× *g* and 4 °C and the pellet was discarded. For BTV, PPRV, and LSDV, the cell association of the viral particles was exploited to further concentrate the virus preparations. For this, the culture medium from several culture flasks was combined in a 50-mL conical tube and centrifuged for 15 min at 3000× *g* and 4 °C. The supernatant was discarded and the cell pellet was resuspended in 2 mL of MEM. All master stocks were aliquoted and stored at −80 °C. To determine their titers, one aliquot of each was thawed and titrated on the cells listed in Table 1. In addition, the viral genome loads of the stocks were defined by quantitative real-time PCR or RT-PCR. 

Tissue culture plates (Corning) for the inactivation experiments (24-well plates) and transfections (6-well plates) were prepared one day in advance by seeding cells in medium with 10% FBS to obtain approximately 80% confluent monolayers on the day of inoculation. Before the cells were infected, the culture medium was replaced with fresh serum-free medium. 

### 2.2. Cards and Filters 

A total of six BCCs and two standard filter papers supplied by Ahlstrom-Munksjö (Helsinki, Finland), Whatman (Cytiva), Copan (Brescia, Italy), Macherey-Nagel (Düren, Germany), and VWR (Radnor, PA, USA) were used for the study. A detailed list of the manufacturers, products and catalogue numbers can be found in Table 2. The cards can be used with a multitude of biological samples [50,51,52,53] except for the Nucleic-Card, which is designed for blood and buccal cells only, and the NucleoCard by Macherey-Nagel, which is intended exclusively for blood samples [54]. Samples spotted on the cards can subsequently be used for a variety of genome analyses, although only the FTA Classic card from Whatman explicitly mentions diagnostics [51]. Copan refers to forensic use [53] and Ahlstrom-Munksjö additionally to biobanking [50], while the intended purpose given by Macherey-Nagel is long-term storage and subsequent examination by real-time PCR [54]. Only Whatman mentioned the stabilization of “nucleic acids” in general [51], while Ahlstrom-Munksjö [50], Copan [53], and Macherey-Nagel [54] refer exclusively to DNA. Nucleic acid stabilization is claimed for all cards [50,51,53,54] except the Human ID Bloodstain Card, whose untreated matrix was not designed for long-term storage [52]. In the case of GenSaver 2.0 [55], the FTA Classic card by Whatman [56] and Copan’s Nucleic-Card [53], the prevention of microbiological growth during the long-term storage at ambient temperature is asserted. All cards in the study feature four circular collection areas with a diameter of 2.5 cm, secured by a flip-down protective cover. The filters are standard uncoated laboratory filter papers, intended for the separation of fine solid particles from liquids.

### 2.3. Inoculation and Sample Collection 

For each sample collection area of the cards, 125 µL sample material was spotted in the center. On the filter paper discs, collection areas of identical size to those of the cards had to be marked first. After spotting, cards and filter papers were air-dried in a biological safety cabinet (BSC) for at least 3 h. The dry inoculated cards and papers were placed in plastic bags inside sealed containers and kept at room temperature in the dark. Samples were recovered from the cards and papers after one day, one week, and one month, respectively. A quarter of each spot (which corresponds to 31.25 µL of applied sample) was cut out using sterile scissors and forceps and put in a 2 mL screw-cap tube with 1750 µL serum-free medium with antibiotics (1% Antibiotic-Antimycotic 100×, Thermo Fisher Scientific, Darmstadt, Germany; 0.4% Gentamicin/Amphotericin B 500×, Thermo Fisher Scientific) and a 4 mm stainless steel bead. The samples were macerated in a TissueLyser II (Qiagen, Hilden, Germany) for 3 min at 30 shakes/s. To separate the supernatant, the tubes were centrifuged at 3000× *g* for 5 min. About 150 µL of liquid was retained in the card matrix and 1600 µL was recovered for further experiments.

### 2.4. Inactivation Experiments

#### 2.4.1. Preliminary Tests: Cytotoxicity and pH

The cards and filters were inoculated with serum-free medium instead of virus culture supernatant and then prepared as described above. Monolayers of all four cell lines were inoculated with 100 µL of a dilution series of the supernatant from the macerated card and filter material down to a 10^−6^ dilution. After incubation for 24 h at 37 °C and 5% CO_2_, cytotoxic effects were evaluated under the microscope. For the GenSaver 2.0 card and Vero dog SLAM (VDS) cells, the experiment was repeated with smaller dilution steps (1:10, 1:20, 1:30, etc., down to 1:100).

To determine the pH value of the card and filter macerates, one quarter of each card was processed as described above, but in distilled water instead of cell culture medium. The supernatant was then dripped onto a pH indicator strip (MQuant, Merck, Darmstadt, Germany) covering a range of pH values from 0 to 14. As a neutral control, the pH of the distilled water was also analyzed. The evaluation was performed visually based on the color change of the strip. 

#### 2.4.2. Virus Inactivation 

The macerates of the spotted FTA Classic card and Nucleic-Card were diluted 1:10 in serum-free medium with antibiotics and then inoculated into cultures of all four cell lines. The same was carried out for the samples from GenSaver 2.0 cards, but the 1:10 dilution was only used for BHK-21, LFBK-αVβ6, and MDBK cells. For VDS cells, the pre-dilution was 1:20. All dilutions were taken into account to calculate the limits of virus detection and inactivation. No dilution was necessary for the remaining cards and the filters.

For each combination of card/filter and virus, 500 µL of the (diluted) macerates were added to 3 wells of a 24-well cell culture plate. The cell line and incubation time at 37 °C and 5% CO_2_ was virus-specific (see above). The cultures were examined for CPE with a microscope and the plates were then frozen at −80 °C in sealed containers. The cells were lysed by freezing and thawing to recover the entire contents of the wells for a second culture passage. For the passage, 500 µL of the lysate were transferred to one day old cell monolayers in medium containing 5% FBS. Again, CPE was assessed after incubation at 37 °C and 5% CO_2_. On each plate, two wells containing only cells in medium were used as a negative control and two wells with cells and 500 µL virus culture supernatant, diluted in the same way as the samples, were used as a positive control. 

The final evaluation was carried out on a percentage basis. It was calculated how many of the tested replicates, per card, virus and time point, caused CPE in culture. For BTV, FMDV, and LSDV, 48 replicates per card or filter and time point were tested. For PPRV, there were 36 replicates. 

### 2.5. Nucleic Acid Stabilization and Extraction

Nucleic acid stabilization for all viruses was assessed by quantitative real-time PCR or RT-PCR. The supernatants of the card and filter macerates were extracted using the NucleoMagVET kit (Macherey-Nagel) on the KingFisher Flex platform (Thermo Fisher Scientific) according to the manufacturers’ instructions. A positive extraction control containing the original virus cell culture supernatant was included in the extraction. 

Previously published assays were used for the viral genome quantification: Hofmann et al. for BTV [57], Callahan et al. for FMDV [58], Batten et al. for PPRV [59], and Bowden et al. [60] with a modified TaqMan probe according to Dietze et al. [61] in the case of LSDV. A heterologous internal control was included in all runs [62]. 

For the RNA viruses BTV, FMDV, and PPRV, the AgPath-ID One-Step RT-PCR Reagents (Thermo Fisher Scientific) were used for the RT-PCR. Here, the total reaction volume was 12.5 µL, including 1.25 µL nuclease-free water, 6.25 µL 2× RT-PCR buffer, 0.5 µL 25× RT-PCR enzyme mix, 1 µL primer-probe mix for the virus detection (with 7.5 pmol of each primer and 2.5 pmol of FAM-labeled probe), 1 µL EGFP primer-probe mix 4 for the internal control (with 2.5 pmol of each primer and 2.5 pmol of HEX-labeled probe), and 2.5 µL template RNA. Cycling conditions were as follows: 10 min at 45 °C, 10 min at 95 °C followed by 42 cycles of 15 s at 95 °C, 20 s at 56 °C, and 30 s at 72 °C. To improve primer binding to the double-stranded genomic RNA of BTV, the template RNA in the PCR plates was denatured at 95 °C for 5 min in a heating block, followed by snap freezing in liquid nitrogen. The master mix was then added to the frozen RNA. For LSDV, the QuantiTect Multiplex PCR NoROX Kit (Qiagen) was used with a total reaction volume of 12.5 µL containing 1.75 µL nuclease-free water, 6.25 µL 2× QuantiTect Multiplex PCR NoROX Master Mix, 1 µL primer-probe mix for virus detection, 1 µL EGFP mix 4 and 2.5 µL template DNA. The temperature profile used was 15 min at 95°C, followed by 42 cycles of 60 s at 95 °C, 30 s at 60 °C, and 30 s at 72 °C.

Samples with a cycle of quantification (C_q_) value > 40 were defined as negative. All assays were run in duplicate with the two resulting C_q_ values subsequently averaged. For each sample, a delta C_q_ value (ΔC_q_) was determined. This was done by subtracting the averaged C_q_ value of the sample from the C_q_ value of the appropriate positive control (per card or filter type, virus and time point). Since several repetitions were performed, the mean value of the individual ΔC_q_ values was then determined for each sample. This corresponds to the increase in C_q_ value (i.e., decreased detection of viral nucleic acid) that results from applying the sample to the card or filter and re-extracting the nucleic acid, compared with analyzing the sample directly. 

For BTV, FMDV, and LSDV, 20 replicates per card or filter and time point were tested. For PPRV, there were 18 replicates.

A virus-specific mean ΔC_q_ across all cards and filter papers was calculated to see which virus was stabilized and/or recovered most effectively. Finally, the average overall loss was calculated by averaging all ΔC_q_ values regardless of the card or filter type, time point, or virus used. The same was conducted with only the best-performing card types.

To evaluate the statistical significance of the results, the 95% confidence intervals of the mean ΔC_q_ values were calculated for each virus, card, or filter and time point using the following formula, where $\bar{x}$ stands for the sample mean, σ for the standard deviation, and n for the sample size:x¯±1.96×σ√n

The calculations were conducted with Microsoft Excel or with GraphPad PRISM for the visual representations, respectively. Non-overlapping confidence intervals were considered evidence of a statistically significant difference. 

### 2.6. Extraction and Transfection of FMDV Samples

Virus culture supernatant of FMDV strains A_22_/IRQ/24/64, A/IRN/08/2015, and O/FRA/1/2001 as well as homogenized epithelial tissue from tongue lesions were used for the transfection experiments. Pieces of vesicular epithelium (approximately 3 mm × 3 mm) from a bovine infected with FMDV A_22_/IRQ/24/64 were homogenized in 400 µL serum-free DMEM, using a 4 mm stainless steel bead in the TissueLyser II (Qiagen) for 3 min at 30 shakes/s. After centrifugation at 3000× *g* for 5 min, the supernatant was obtained. 

The cell culture and tissue supernatants were used to prepare a 10-fold dilution series in serum-free DMEM down to a dilution of 10^−4^. A volume of 125 µL of the original supernatant and the 10^−1^ dilution were spotted on each card or filter. The tubes with the dilution series were stored at −80°C. The cards and filters were dried overnight in a BSC. The following day the spots were macerated as described above, but using only 400 µL of serum-free DMEM. RNA was extracted from the supernatants using TRIzol LS reagent (Thermo Fisher Scientific), which is an improved single-step RNA isolation method originally described by Chomczynski and Sacchi [63]. It was performed following the manufacturer’s instructions. The RNA pellet was dried on a heating block for approximately 5 min at 30 °C. The contents of the reaction tubes used to spot the cards, as well as the previously unused 10^−3^ and 10^−4^ dilutions, were thawed and treated in the same way as the samples for use as a positive control. The extracted RNA was then immediately used for transfection. 

Lipofectamine 3000 (Thermo Fisher Scientific) was used for the transfections. For each preparation, 1 µL of Lipofectamine 3000 reagent was diluted in 25 µL of serum-free DMEM and mixed gently without vortexing. In a second tube, 3 µL of RNA, 0.5 µL P3000 reagent and 25 µL DMEM were mixed in the same way. The two preparations were then combined dropwise and incubated for 10 min at room temperature. Fifty µL of the mixture was added to one well of a 6-well cell culture plate, again dropwise and without touching the surface. The following incubation of 15 min was carried out at 37 °C and 5% CO_2_. Finally, the wells were filled with 500 µL DMEM with 5% FBS and incubated at 37 °C and 5% CO_2_ for two days. Two wells were used per sample. The remaining volume of eluate was used for real-time RT-PCR and subsequently stored at −80 °C. After the incubation, the monolayers were checked for signs of CPE under the microscope and the plates were then frozen at −80 °C. A passage and second reading of CPE was performed as described above. These tests were repeated at least two times for each virus strain. 

### 2.7. Sequencing of FMDV Samples

RNA samples from cards that could not be successfully transfected were used for the sequencing analyses. The qScript XLT One-Step RT-PCR ToughMix (Quantabio, Beverly, MA, USA) and the previously published primers FMD-3161-F and FMD-4303-R were used to amplify the VP3- and VP1-coding regions of the FMDV genome with an amplicon size of 1143 bp [64]. The total reaction volume of 10 µL contained 0.6 µL of nuclease-free water, 5.0 µL 2× One-step Tough Mix, 0.4 µL qScript XLT OneSTep RT enzyme, 1 µL each of the forward and reverse primer at a concentration of 10 pmol/µL, and 2.0 µL template RNA. The temperature profile was 20 min at 48 °C, 3 min at 94 °C, followed by 40 cycles of 15 s at 94 °C, 30 s at 60 °C, 60 s at 68 °C, and finally 5 min at 68 °C. The PCR products were then run in an agarose gel. Bands of the expected size were excised and DNA was extracted with the QIAquick gel extraction kit (Qiagen) and sent to Eurofins Genomics (Ebersberg, Germany) for sequencing.

## 3. Results

### 3.1. Virus Propagation 

The same master stock of each virus was used for all experiments. The titer of the BTV master stock was 10^8.0^ TCID_50_/mL and it had a C_q_ value of 7.9, FMDV A_22_ Iraq was 10^7.8^ TCID_50_/mL and C_q_ 12.5, FMDV A/IRN8/2015 was 10^7.3^ TCID_50_/mL and C_q_ 12.3, FMDV O/FRA/1/2001 was 10^5.3^ TCID_50_/mL and C_q_ 14.4, PPRV Nigeria 75/1 was 10^6.5^ TCID_50_/mL and C_q_ 13.3, and LSDV Neething V100 was 10^7.7^ TCID_50_/mL and C_q_ 13.7.

### 3.2. Inactivation Experiments

#### 3.2.1. Cytotoxicity

No cytotoxicity was observed for the GenSaver, Human ID Bloodstain Card and NucleoCard macerates in any cell line. In contrast, all cell lines inoculated with the undiluted card macerates from GenSaver 2.0, FTA Classic and Nucleic-Card showed obvious signs of cytotoxicity. The cytotoxic effects were abolished when the macerates were diluted 1:10 or, in the case of GenSaver 2.0 and VDS cells, 1:20. The cytotoxicity data for all cards and filters are summarized in Appendix A.

#### 3.2.2. pH

The pH of the macerates of the Human ID Bloodstain card, NucleoCard, and the filter papers was determined to be approximately 5, the same as the pH of the distilled water. For GenSaver, GenSaver 2.0, FTA Classic, and the Nucleic-Card the pH was approximately 7. FMDV is particularly sensitive to even mildly acidic pH [65].

#### 3.2.3. Virus Inactivation

Taking into account the dilutions required to avoid cytotoxicity, the main experiment demonstrated complete inactivation of at least 10^4^ TCID_50_ spotted on the card for all cards and filters and all viruses except PPRV (Figure 1). Due to the lower initial titer of the PPRV master stock, inactivation of a virus dose of 10^4^ TCID_50_ could only be shown for cards where no dilution of the macerate was required (GenSaver, Human ID Bloodstain Card, NucleoCard). For the samples that had to be pre-diluted, i.e., 1:10 for FTA Classic and Nucleic-Card and 1:20 for GenSaver 2.0, the inoculated virus dose (which was completely inactivated) was 10^3.7^ TCID_50_ and 10^3.6^ TCID_50_, respectively. 

In summary, almost all cards or filters markedly reduced virus infectivity over time for almost all viruses. PPRV was rapidly inactivated by all cards and filters within less than one day. For BTV and FMDV, there often was residual infectivity after one day of incubation and the largest decrease in infectivity was seen between one and seven days. GenSaver 2.0, FTA Classic and Nucleic-Card inactivated BTV in less than one day and NucleoCard within one week of incubation. For GenSaver, Human ID Bloodstain Card and the Whatman filter paper, complete inactivation of BTV was demonstrated after 30 days of incubation. On the VWR filter paper, BTV remained infectious virus even after one month. FMDV was inactivated by the Human ID Bloodstain Card and the Whatman filter paper within less than a day. All other cards completely inactivated FMDV within a week, except for NucleoCard and the VWR filter paper, which required a month of incubation for full inactivation. The inactivation of LSDV was most challenging overall. While GenSaver 2.0, FTA Classic and Nucleic-Card completely inactivated LSDV within less than a day, this was not possible for GenSaver, Human ID Bloodstain Card, NucleoCard and the filter papers. They were unable to completely inactivate LSDV even after one month of incubation, and GenSaver did not show a reduction of infectivity at all (Figure 1). 

### 3.3. Nucleic Acid Stabilization

Clear differences were also observed in the stabilization efficacy (Figure 2). On one hand, the stability of nucleic acids on a card or filter can be observed over time, demonstrating that the level of detectable virus genome remained approximately the same over time for all cards, filters, and viruses used.

On the other hand, the amount of viral RNA or DNA that can be eluted from the cards or filters can be compared to the original liquid sample, revealing a wide range of ∆C_q_ values. Among the specimens with the highest average ∆C_q_ values for all viruses, i.e., the greatest loss of nucleic acid, were the filter papers. The cards with the lowest average ∆C_q_ values, i.e., the best recovery of nucleic acid from all viruses, were GenSaver 2.0, FTA Classic and Nucleic-Card. Their mean ∆Cq value was 4.6, whereas the mean ∆Cq value for all cards and filters was 6.3. Between virus groups, the best recovery was seen for the double-stranded DNA genome of LSDV, with a mean of all ∆C_q_ values of 5.6, followed by FMDV with 5.8 and BTV with 6.6. The highest loss was found for PPRV with a mean ∆C_q_ value of 7.1. 

The significance of the results was evaluated by calculating the 95% confidence intervals of the mean ∆C_q_ values. The widely overlapping confidence intervals indicated that there were no significant differences between the cards and filters in the stabilization of BTV. For FMDV, PPRV, and LSDV, on the other hand, the cards A2, W and C showed significantly better results, especially in comparison with the filter papers FW and FV. 

### 3.4. Transfection 

Successful recovery of infectious FMDV from spots of virus culture supernatant was possible for GenSaver 2.0, the FTA Classic card, Nucleic Card, and NucleoCard (Table 3). Across all experiments, transfection was successful up to a maximum C_q_ of 27.0 for card or filter samples and 32.6 for positive controls (original liquid samples). Detailed information is presented in Appendix A.

For the epithelial homogenate from a tongue lesion of an animal infected with FMDV A_22_ Iraq, the C_q_ value of the RNA extracted directly from the undiluted liquid sample was 14.8. This was the only RNA that could be successfully transfected. No infectious virus was recovered from the RNA extracted from higher dilutions of the liquid sample (with C_q_ values of 18.6, 23.1, 27.3, and 31.2) nor from any of the RNA extracted from cards spotted with the epithelial homogenate. 

### 3.5. Sequencing 

Failure to recover infectious virus by transfection indicates the absence of intact full-length viral genomes. For these samples, it was analyzed whether the fragmentation of the RNA still allowed PCR amplification and sequencing. This did not reveal any differences between the cards. The 1143 bp fragment of the VP3- and VP1-coding regions was successfully amplified from all eluates (see Appendix A). The sequences obtained from the cards matched the previously determined sequences of the used virus strains. 

## 4. Discussion

### 4.1. Sample Collection and Extraction

Biopsy punches are often used to excise card material for nucleic acid extraction [26,42,66,67,68]. Cross-contamination is not considered a problem for this method [67] since the card fibers do not adhere to the punch [36]. Nevertheless, using a “clean” punch [36] or disinfecting the punch between samples is recommended [7,23]. It has not been evaluated whether residual disinfectant can affect subsequent samples. In our experience, the punched-out discs are awkward to manipulate and are so light that they are often blown away by the laminar air flow in the biosafety cabinet when trying to transfer them to reaction tubes. Therefore, we decided to use a fresh set of reusable sterile scissors and forceps for each sample, as described by several previous publications [6,11,14,21].

### 4.2. Inactivation and Stabilization 

Clear differences were seen between the various cards and filters in terms of their inactivation and stabilization efficacy. GenSaver 2.0, FTA Classic, and Nucleic-Card delivered the best results. They were able to inactivate almost all tested viruses within one day at a spotted virus dose of at least 10^4^ TCID_50_, except for the FTA Classic card requiring one week of incubation for the inactivation of FMDV. The critical dose of 10^4^ TCID_50_ is based on the guidelines for testing virucidal disinfectants, where a reduction of infectivity by 4 log_10_ (or 99.99%) is required [69]. The complete inactivation of a virus dose of 10^3.6^ TCID_50_ (as demonstrated for PPRV) still represents a reduction of 99.975%. All tested cards and filter papers inactivated PPRV within less than a day, which is in line with the expected low tenacity of enveloped single-stranded negative-sense RNA viruses [46]. Since the sample with the shortest incubation time was taken after 24 h, no statement can be made about how quickly the cards and filters can actually inactivate PPRV. Further tests with shorter incubation times could be carried out to determine this. When using the other cards for FMDV or BTV, an incubation period of at least a month is required to ensure complete inactivation. Careful selection of the card to use is especially important with DNA viruses such as LSDV, which are much more stable than RNA viruses in samples spotted on storage cards [26]. 

Most cards or filters were able to keep RNA and DNA stable over a long period of time. The recovery of nucleic acid from the cards was most effective for the DNA virus, which gave generally low ∆Cq values compared with the original sample. For the RNA viruses, a considerably smaller fraction of the viral nucleic acid was recovered from the cards, confirming the findings of earlier studies [66]. The lower stability of RNA is also cited as a reason for this. Overall, the cards with the best inactivation efficacy, GenSaver 2.0, FTA Classic and Nucleic-Card, also performed well in the stabilization experiments.

For the positive control, 100 µL of pure virus culture supernatant was used. This corresponds to more material than would be present in a quarter of the spot on a card or filter. It is intended to reflect the best case, where the original liquid sample can be tested directly. In contrast, the cards or filter papers reflect the situation in the field, where the sample must be preserved for storage or transport to the laboratory. 

In summary, the cards that prevent microbiological growth also produce the best inactivation and stabilization results. We can only speculate about possible additives that make the inactivation and stabilization so much better, as the companies do not provide precise information about the formulation of the chemical coating. For this reason, we have been reluctant to write too much about the causes of the very different results for the cards, because we are simply not aware of them.

If the samples eluted from these cards are to be used in cell culture, however, they must be diluted to avoid damage to the cells. Remarkable are the results of the plain filter papers, which did reasonably well in both experiments. This shows that the drying on absorbent paper alone already has considerable inactivating and stabilizing effects [3,4,5,24,70] which are entirely incidental to the intended application of these products.

It should be noted that in this study, with the exception of the FMDV-positive tongue lesion, we used only cell culture supernatant as a sample material. This, of course, does not correspond to sampling in practice. It remains unclear to what extent the results can be transferred to other sample matrices. We assume that this is possible for aqueous samples with low protein content (e.g., saliva). However, for sample materials with high protein content (e.g., whole blood), it is conceivable that the inactivation and stabilization performance may be impaired.

### 4.3. Transfection and Sequencing of FMDV Genomes

A similarly heterogeneous picture emerged from the attempts to recover intact viral genomes from the cards and filters. Transfection was only attempted with RNA eluted one day after spotting on the cards. With GenSaver 2.0, FTA Classic, and Nucleic-Card, transfection of the 10^−1^ dilution was successful for FMDV A/IRN/8/2015 and A_22_ Iraq, but FMDV O/FRA/1/2001 was only successfully transfected from the undiluted RNA. One possible reason for this is the over 100-fold lower initial titer of the latter, which conceivably corresponds to a lower number of full-length viral genomes. 

In summary, with high-titer samples and the right card, a sufficient number of full-length genomes can be recovered to allow successful transfection. Once again, GenSaver 2.0, FTA Classic, and Nucleic-Card performed better than the other cards. To obtain optimal results, sample processing and transfection should be carried out as quickly as possible after the arrival of the cards in the laboratory [38]. PCR amplification and sequencing of the VP1-coding region, on the other hand, was possible with all cards and filter papers. In our study, this was only attempted with cards spotted the previous day, but it has been reported that sequencing was still possible after storing cards at temperatures of 41 °C for 15 days [38]. 

### 4.4. Use of Biosample Collection Cards in Diagnostics

Improper collection or storage of liquid samples can quickly lead to degradation of nucleic acid or overgrowth by microorganisms [71,72], potentially compromising the diagnostic value of the sample [12]. Biosample collection cards offer an inexpensive solution for collecting and sending samples that is widely available and requires little preparation. Due to the ease of shipping via regular mail without refrigeration or strict biosafety requirements [13,23,24], the cards have a clear advantage in terms of logistical effort during sample collection as well as during transport and subsequent storage. Especially in situations with limited resources and infrastructure, this can result in a considerable increase in sample quality and quantity. Especially for large-scale mass surveys, the cards can be used advantageously [2]. Due to the small amount of samples needed, puncturing an ear vein is often sufficient for sampling [73]. This saves materials and time and makes handling animals much less stressful. Additionally, no additional laboratory equipment is required to process the collected samples before shipment. 

It must be noted, however, that the inactivating effect of the cards is restricted to the collection areas intended for the samples. Contamination of the protective envelope must be avoided. As demonstrated in our study, DNA viruses in particular will not be inactivated by contact with absorbent paper alone. 

However, if the sample is applied properly, the risk of transmission during transport is minimized by three levels. The fibers of the card spots immobilize the sample after initial drying [16]. This represents the first level. The flip-down protective cover can be considered the second level. Finally, the sample is shipped in a sealed envelope and is again shielded from the environment, which represents the third level [74].

The biological risk can be further reduced by heating spotted cards to 70 °C for 30 min [75], immersing them in 0.2% acetic acid for 15 min [76], or by treatment with phenol [77]. It is assumed that this will not affect downstream real-time PCR analysis or sequencing, but may cause increased degradation of full-length genomes. Further studies are necessary to determine the diagnostic utility of this approach.

## 5. Conclusions

The performance of the tested cards varied considerably, with GenSaver 2.0, FTA Classic, and Nucleic-Card delivering the best results for inactivation, stabilization, and nucleic acid recovery. 

## Figures and Tables

**Figure 1 viruses-14-02392-f001:**
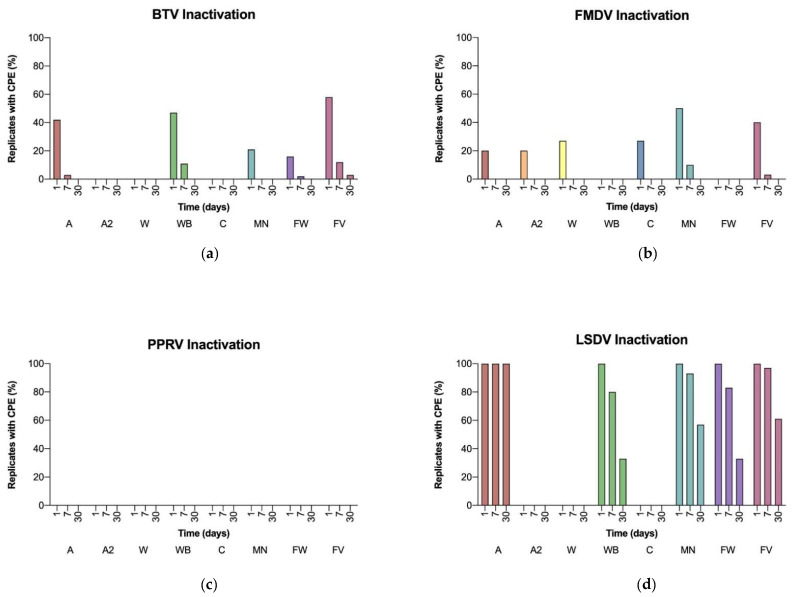
Virus inactivation on cards and filters during one day, one week, and one month of incubation. The percentage of all replicates still causing CPE in cell culture is shown for each virus, card, or filter and time. (**a**) BTV; (**b**) FMDV; (**c**) PPRV; (**d**) LSDV.

**Figure 2 viruses-14-02392-f002:**
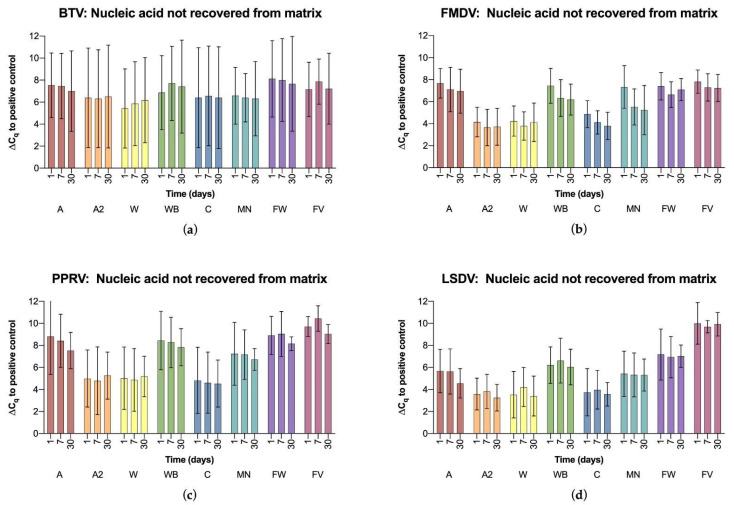
Stabilization and recovery of nucleic acid from cards and filters analyzed by real-time (RT)-PCR after one day, one week and one month of incubation, showing the mean ∆C_q_ values for each virus, sample time point, and card with their 95% confidence intervals indicated by the error bars. ∆C_q_ values were calculated between the eluted nucleic acid and the virus preparation used for spotting (positive control). (**a**) BTV; (**b**) FMDV; (**c**) PPRV; (**d**) LSDV.

**Table 1 viruses-14-02392-t001:** Cell lines and media used for virus culture.

Virus	Cell Line	FLI Cell Culture Collection No.	Culture Medium	Cell Count for Seeding ^3^
BTV	BSR/5	RIE0194	MEM ^1^	1.25 × 10^5^
FMDV	BHK-21 clone Tübingen	RIE0164	MEM	1.25 × 10^5^
FMDV	LFBK-αVβ6 [48]	RIE1419	DMEM ^2^	1.5 × 10^5^
PPRV	Vero dog SLAM [49]	RIE1280	MEM	1.25 × 10^5^
LSDV	MDBK	RIE0261	MEM	1.0 × 10^5^

^1^ Minimal essential medium with Hanks’ and Earle’s salts and non-essential amino acids. ^2^ Dulbecco’s modified Eagle medium. ^3^ Number of cells seeded per well of a 24-well culture plate to obtain an 80% confluent monolayer after 24 h. Cells in plates were incubated at 37 °C with 5% CO_2_.

**Table 2 viruses-14-02392-t002:** Biosample collection cards and filter papers used in the study.

Manufacturer	Product Name (Catalogue Number)	Code	Sample Type	Application	Special Features
Ahlstrom- Munksjö	GenSaver (8.564.0002.B-N)	A	Biological fluids	**DNA:** Forensic, Biobanking markets	
Ahlstrom- Munksjö	GenSaver 2.0 (8.566.0002.B-N)	A2	Biological fluids	**DNA:** Forensic, Biobanking markets	Prevents growth of microorganisms
Whatman	FTA Classic Card (WB120205)	W	Biological samples	**Nucleic acid:** Diagnostic, clinical applications, bacterial, viral, blood, plant, insect analysis, Genomic, Forensic...	Prevents growth of microorganisms
Whatman	Human ID Bloodstain Card (WB100014)	WB	Blood, bodily fluids	**DNA:** Short-term handling (collection and transport)	
Copan Flock Technologies	Nucleic-Card (4473977)	C	Blood, buccal cells	**DNA:** Forensic	Prevents growth of microorganisms
Macherey- Nagel	NucleoCard (740403.10)	MN	Blood	**DNA:** Long term storage, Real-time PCR analysis	
Whatman	Grade 1 filter paper (1001-110)	FV		Filtration	
VWR	Grade 413 filter paper (516-0815)	FW		Filtration	

**Table 3 viruses-14-02392-t003:** Transfection results by card or filter type, virus strain and sample dilution, showing the number of successful transfections compared to the total number of replicates. The average C_q_ value of the samples for which the transfection was successful is given in parentheses.

Product Name	Virus Strain and Dilution					
	FMDV A_22_ Iraq		FMDV A/IRN/8/2015		FMDV O/FRA/1/2001	
	Undiluted	10^−1^	Undiluted	10^−1^	Undiluted	10^−1^
GenSaver	0/3	0/3	0/2	0/2	0/2	0/2
GenSaver 2.0	1/3 (21.2)	1/3 (26.7)	2/2 (18.5)	1/2 (22.9)	2/2 (20.5)	0/2
FTA Classic card	1/3 (21.2)	1/3 (25.8)	2/2 (20.0)	1/2 (24.2)	2/2 (21.8)	0/2
Human ID Bloodstain Card	0/3	0/3	(0/2)	(0/2)	0/2	0/2
Nucleic-Card	2/3 (21.0)	1/3 (24.7)	2/2 (19.6)	1/2 (26.8)	1/2 (24.2)	0/2
NucleoCard	2/3 (20.9)	0/3	1/2 (19.0)	0/2	0/2	0/2
Filter paper, Whatman	0/3	0/3	0/2	0/2	0/2	0/2
Filter paper, VWR	0/3	0/3	0/2	0/2	0/2	0/2

## Data Availability

Summarized data from the study are included in the manuscript and the Appendix A. Raw data for the individual experiments will be provided by the authors upon justified request.

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
