# Peer review of "Comparison of Biosafety and Diagnostic Utility of Biosample Collection Cards"

_viruses, 2022, doi:10.3390/v14112392_

Round 1

Reviewer 1 Report

The article compared 6 bio-sample collection cards, and showed their ability to inactivate viruses and stabilize viral nucleic acid. The result showed that most cards could keep RNA and DNA stable for a long period of time, which gave us a good guidance on their utility. However, for the results of this manuscript, I have some major suggestions.

1. How about the retention time if the viral nucleic acid concentration is low? We suggest the author compare the stability at more dilutions, and to see if there are some findings.

2. To detect the cytotoxicity, it is inaccurate to conduct morphology observation by microscope. CCK8 or MTT assay should be performed.

3. The author chose different types of RNA viruses and DNA viruses, but I found that only different serotypes of FMDV were selected. Does this make sense? Whether different strains of other viruses could be selected to verify the reliability of the results?

4. From the results of Figure 1, we can find that almost all cards and filters have a good inactivation effect on BTV, FMDV and PPRV, especially PPRV. But the results is too significant, and the experimental conditions should be improved, such as shortening the time of inactivation on the PPRV virus makes the results more realistic.

5. There were only A2, W and C have a significant inactivation effect on the inactivation of LSDV, while other cards and filters have no significant inactivation effect. It should be analyzed and speculated the possible reasons for this result in the discussion section.

6. In addition to detecting the Cq value of the virus by RT-PCR, the standard for each virus should also be prepared, and a standard curve should also draw. Convert the Cq value of each virus into the number of virus copies through standard curve, instead of directly using the virus Cq value.

7. In Figure 1, through the results of the TCID50, we found that the CPE after virus infection showed significant differences in BTV, FMDV and PPRV. But the difference in the Cq value of viral nucleic acid was not significant, especially for PPRV. Why are the results of the two experiments so different?

Besides, minor suggestions and corrections for the author.

1. In line 118, it’s better for the sentence “The supernatant was decanted and discarded” changed to “The supernatant was discarded.”

2. For the material section of cards and filters, it’s better to add more columns in table 2 to illustrate more information, people would learn the difference between products clearly.

3. In 3.1, the authors provide results for 6 viruses, but in table1, the authors only list information for 5 viruses, please complete the list.

4. Line 265, fifty μL should be revised to 50 μL.

5. μl should be revised toμL. Please also correct similar mistakes in the manuscript.

6. Line 309, 104 TCID50 should be revised to 104 TCID50/mL or 104 TCID50/0.1 mL. Please also correct similar mistakes in the manuscript.

7. Please provide statistical analysis results for the data in the manuscript.

Reviewer 2 Report

The authors present data on 6 different bio-sample collection cards and 2 different filter papers loaded with cell culture supernatant for several important livestock viral diseases. Preliminary tests included assessment of cytotoxicity and pH of the cards, followed by virus inactivation, nucleic acid stabilization and extraction, and transfection for FMDV samples. The only ex vivo sample included, was homogenized tongue lesions used for FMDV transfection experiments.

Major comments

I would question if the authors have truly tested the diagnostic utility of the collection cards, and if the data presented reflect the title. Additional data could be provided, or the discussion section could be expanded to address the following points:

·         Data is presented for samples loaded with cell culture supernatant. How do these results translate, or not translate, to routine diagnostic samples collected from infected animals for biosafety and diagnostic utility. Would the sample matrix have a significant impact on the biosafety and diagnostic potential presented?

·         Linked to the point above, the authors report data for tongue lesion homogenates under 3.4. The authors speculate under 4.3, that homogenization could have fragmented the viral genome, and led to lower recovery from ex vivo samples. Could the sample matrix have impacted these results? Additional experiments could be done: for example, homogenize cell culture supernatant as a control, perform TCID50 and standardize the sample input, spike the vesicular material etc..

·          For the data presented in figure 2, can the authors confirm which results are significant?

·         The discussion section is results focused, and could benefit from an overview of how the work is relevant for the field for diagnostic and biosafety utility. Interpretation of the data could also help the reader, for example: in figure 1, the difference between GenSaver (A) and GenSaver 2.0 (A2) for LSDV inactivation is striking, but not discussed in detail. The conclusion could be expanded.

Minor comments   

·         Line 17 and 18 of the abstract could be improved (‘the inactivation of the RNA viruses did not pose any problems…)

·         In figure 1, is it necessary to show the blank panel for PPRV inactivation?

Round 2

Reviewer 2 Report

No additional comments.

Author Response

Comments for authors from Academic Editor:
"Author's reply to referee 1 comments are in the most part ok, however, even with the introduction of Supplementary table 4 that shows confidence intervals, I think that it would be better if authors add the statistical results in the figures. In my opinion, it would improve MS quality and would be recommended for a Viruses paper. "

>According the recommendation of the academic editor the confidence intervals were included in the figures.